# Fluoride Evaporation of Low-Fluoride CaF_2_-CaO-Al_2_O_3_-MgO-TiO_2_-(Na_2_O-K_2_O) Slag for Electroslag Remelting

**DOI:** 10.3390/ma16072777

**Published:** 2023-03-30

**Authors:** Bo An, Yue Gu, Jiantao Ju, Kun He

**Affiliations:** 1Shaanxi Steel Group Hanzhong Iron and Steel Co., Ltd., Hanzhong 723000, China; 2School of Metallurgical Engineering, Xi’an University of Architecture and Technology, No.13 Yanta Road, Beilin District, Xi’an 710055, China; 3Research Center of Metallurgical Engineering Technology of Shaanxi Province, Xi’an 710055, China

**Keywords:** electroslag remelting, fluoride evaporation, thermogravimetric analysis, structure, thermodynamic calculation

## Abstract

To elucidate the behavior of fluoride evaporation in an electroslag remelting process, the non-isothermal evaporation of the low-fluoride CaF_2_-CaO-Al_2_O_3_-MgO-TiO_2_-(Na_2_O-K_2_O) slag is studied using thermogravimetric analysis. The evaporation law of the melted slag is further verified using thermodynamic calculations. Fourier transformation infrared (FTIR) spectroscopy is used to evaluate the change in slag structure. It is discovered that the principal evaporating substances are CaF_2_, KF, and NaF, while the evaporation of MgF_2_, AlF_3_, and AlOF is less. KF evaporates absolutely in the early stage of the reaction, and CaF_2_ evaporates in a large proportion during the late reaction period. At 1500 °C, the order of vapor pressure is KF > CaF_2_. When K_2_O and Na_2_O are added to the residue sample at the same time, the evaporation ability of KF is stronger than that of CaF_2_ and NaF. As the K_2_O content increases from 0 to 8.3 wt%, evaporation increases from 0.76% to 1.21%. The evaporation rates of samples containing more K_2_O and those containing more Na_2_O are 1.48% and 1.32%, respectively. Under the same conditions, K_2_O has a greater effect on evaporation than Na_2_O. FTIR results show that the addition of K_2_O depolymerizes the network structure and that K_2_O can depolymerize the network structure better than Na_2_O.

## 1. Introduction

Electroslag remelting (ESR), a secondary refining technology combining refining and directional solidification, is widely used in the production of high-performance special steel and super alloys [1,2]. Considering the unique and outstanding advantages of ESR, such as uniform composition, effective removal of non-metallic inclusions, significant dephosphorization and desulfurization, and good surface quality, ESR has become an important technology in the metallurgical industry [3,4,5]. Slag plays an important role in the ESR process: (1) it acts as a heat source in the remelting process; (2) leads to efficient refining; (3) has a protective effect on the slag; and (4) during the solidification process of remelted metal, the ingot’s surface develops a thin, homogeneous layer of slag that shields the mold from the high-temperature slag’s direct impact while also making the ingot’s surface smooth and simple to demold [6,7,8]. As the primary component of the electroslag system, the slag’ surface tension and viscosity may be reduced using CaF_2_, and the slag’s conductivity can be increased [9,10]. Around 40–70 wt% of conventional slag is made up of CaF_2_ [11], which is the primary substance involved in fluoride evaporation during remelting. The evaporation of fluoride changes the components of the slag system, causing potential harm to health and safety, and gaseous substances released, such as HF, CaF_2_, AlF_3_, AlOF, MgF_2_ and others, cause serious environmental pollution problems [12,13,14,15]. With regard to the removal of fluoride evaporation, there are still some typical challenges to the process, which have often been overlooked. In addition, there is a shortage of CaF_2_ [16], so efforts are made to develop a slag system using oxides instead of fluorides. The development of low-fluorine slag or fluorine-free slag would not only reduce fluorine pollution but also meet the needs of ESR slag system. However, employing low-fluorine slag or fluorine-free slag still has several difficulties, including a high melting point and low electrical conductivity, which leads to some difficulties with ignition, thin slag shell and uneven thickness, which affect the smoothness of the ingot surface, and several other problems [17,18]. Therefore, it is urgent to find an alkaline oxide to replace CaF_2_ use in slag to solve these problems [19].

Previous studies have investigated the behavior of density, surface tension, and viscosity of alkali metal Na_2_O and K_2_O metallurgical slag. Sukenaga et al. [20] indicated that the density and surface tension of the CaO-SiO_2_-Al_2_O_3_ slag decreased with the addition of Na_2_O and K_2_O. Hou et al. [21] proposed that the addition of Na_2_O and K_2_O to the CaO-SiO_2_-Al_2_O_3_ slag reduced the viscosity, and that the degree of reduction is greater for Na_2_O than for K_2_O. Alkali metal oxides show good performance in CaO-SiO_2_-Al_2_O_3_, a fluorine-free slag system. Chang et al. [22] discovered that, at the same temperature, adding K_2_O increased the viscosity of slag while adding Na_2_O lowered it in CaO-MgO-Al_2_O_3_-SiO_2_ slag. Zhang et al. [23] reported that the addition of Na_2_O or K_2_O to CaO-SiO_2_-(K_2_O) melts led to a decrease in conductivity. By replacing Na_2_O with K_2_O, the electrical conductivity showed a tendency to decrease and then increase with the increase of K_2_O substitution. The behavior of alkali metal oxides in some slag systems requires further study. Studies on the addition of alkali metal oxides to ESR slag systems containing CaF_2_ have also been conducted. Zheng et al. [24] investigated the effect of adding alkali metal Li_2_O on evaporation and crystallization of an ESR slag system containing CaF_2_. It was found that as the Li_2_O content increased, the liquid phase line temperature decreased and evaporation increased, while the crystallization behavior was inhibited. It was shown that the appropriate amount of Li_2_O can be used as an effective component for the design of low-fluorine ESR slag. Shi et al. [18] investigated the effect of replacing CaF_2_ with alkali metal Li_2_O on the viscosity and structure of ESR slag systems containing CaF_2_ in order to develop a low-fluorine slag for ESR; the study showed that the increase in Li_2_O reduces the viscosity and structural polymerization of the slag. Due to their comparable physical and chemical characteristics, alkali metals are excellent CaF_2_ replacements and have been added to a variety of slags to modify their properties in order to meet the specifications of the ESR slag system. After experimental verification [25], the addition of Na_2_O to CaF_2_-CaO-Al_2_O_3_-MgO-TiO_2_, the basic slag system, has the effect of lowering the melting point as well as the viscosity of the slag system; this is applicable to the development of the ESR process and is a worthwhile study. Previous studies focused only on the alkali metals Li_2_O and Na_2_O. Therefore, based on previous studies, in this study, K_2_O was added into the ESR slag system containing CaF_2_ to study the influence of alkali metal K_2_O on the evaporation of the ESR CaF_2_-CaO-Al_2_O_3_-MgO-TiO_2_ slag system; the influence of alkali metals Na_2_O and K_2_O was compared.

A new slag system was adopted in this study. The content of CaF_2_ was fixed at 30 wt%, which is a low fluorine slag system that has been under studied in the past. Rather than focus on the more typical characteristics of the slag system, such as melting point, viscosity and electrical conductivity, this study focuses on the evaporation and structure of the slag system. The current study’s objective is to investigate the underlying mechanism by which K_2_O addition affects the evaporation and structural behavior of CaF_2_-CaO-Al_2_O_3_-MgO-TiO_2_-(Na_2_O-K_2_O) slag for ESR. The influence of Na_2_O and K_2_O content on the slag system with low fluorine is also compared when added in the same amount in order to further understand the effect of mixed alkali on the slag.

## 2. Experiment

### 2.1. Materials and Sample Preparation

Reagent-grade powders of CaF_2_ (≥98.5%), CaO (≥98.0%), Al_2_O_3_ (≥99.0%), MgO (≥98.0%), TiO_2_ (≥99.0%), Na_2_CO_3_ (≥99.8%), and K_2_CO_3_ (≥99.0%), supplied by Sinopharm, Inc. (Sinopharm Chemical Reagent Co., Ltd., Shanghai, China), were used in the present experiment. The Na_2_O and K_2_O powders were derived from reagent grade Na_2_CO_3_ and K_2_CO_3_. The CaO powder was calcined at 1000 °C for 4 h in a muffle furnace; the remaining components were dried at 200 °C for 4 h in a drying oven. The chemical constituents of the designed slag samples are listed in Table 1. The slag samples A, K1, K2, K3, K4, and K5 were prepared according to the components in Table 1. The samples were ground in a mortar and pestle for 30 min for better mixing. The slag samples were then pre-melted in a platinum crucible in a muffle furnace under an air atmosphere. The pre-melting temperature was 1500 °C; this temperature was maintained for 5 min. Before drying, crushing, and grinding under 200 meshes for the further use, the samples were quenched in water. Due to its evaporation losses, the composition of the quenched slag may differ from that of the designed samples. X-ray fluorescence spectroscopy (XRF, Rigaku ZSX Primus II, Rigaku Corporation, Tokyo, Japan) was then used to analyze the composition for comparison purposes; the outcomes are listed in Table 1. As can be seen, there are only slight differences between the measured values and the intended composition.

### 2.2. Thermodynamic Calculations

Factsage software (GTT Technologies, Aachen, Germany and Thermfact/CRCT, Montreal, QC, Canada) is a combination of two thermochemical software packages: FACT-Win and ChemSage. For this paper, the equilib module was used to calculate the weight of the evaporated species along with FToxid and FactPS databases. The selected temperature range was 650–1500 °C with an interval of 50 °C and a pressure of 1 atm. To assess the gaseous species evaporating from the slag at various temperatures and to quantify their weights, 100 g of each sample was used as the specimen.

### 2.3. XRD and FTIR Spectra Measurement

The pre-melted slag samples were used for X-ray diffraction (XRD, D8 Advance A25; Bruker AXS, Karlsruhe, Germany). As shown in Figure 1, no significant characteristic peaks were found in the samples, indicating that the slag sample was amorphous. Pre-melted amorphous slag was surveyed using a Fourier transform infrared spectroscope. FTIR spectroscopy (Nicolet iS5; Thermo Fisher Scientific, Waltham, MA, USA) was used to characterize the molten slag structure. Using a KBr detector with a spectral resolution of 4 cm^−1^ and a scan number of 32, FTIR transmission spectra in the 4000–400 cm^−1^ region were captured. Each 2 mg sample was combined with 200 mg of KBr in an agate mortar; the resulting mixture was then pressed into a clear flake measuring 13 mm in diameter.

### 2.4. Simultaneous Thermal Analysis Measurement

The non-isothermal evaporation behavior of the slag was evaluated using a simultaneous thermal analyzer (Netzsch STA449F3, Netzsch Instrument Inc., Selb, Germany). Thermogravimetric analysis (TG) and differential scanning calorimetry (DSC) were performed on the slag. The temperature was determined using a B-index thermocouple under an argon atmosphere with a gas flow rate of 80 mL/min. For each TG-DSC measurement, approximately 50 mg of material was heated in a platinum crucible at a rate of 20 °C/min from 50 °C to 1500 °C for 1 min to remove bubbles and homogenize its chemical makeup. The platinum crucible dimensions were 5.5 mm inner diameter and 6.3 mm high.

## 3. Results and Discussion

### 3.1. Thermodynamic Analysis

#### 3.1.1. Proportion of Fluorinated Gas Evaporation

To clarify the weight of the evaporation components and the proportions of the main fluorine gases released from the pre-melted slag, Factsage 8.0 software was used. It is clear from the calculations that the reaction begins at 650 °C, so the variation in the 650–1500 °C temperature range was studied. These calculated curves are visualized in Figure 2. These curves show the proportions of the main fluoride gases released in the pre-melted slag at different temperatures. The results show that when there is no K_2_O in the slag sample, CaF_2_ is the main evaporative substance, occupying an absolute advantage. When K_2_O is added, CaF_2_ and KF are the main evaporation species. KF begins to evaporate at 650 °C; in the temperature range of 650–900 °C, only the reaction of CaF_2_ and K_2_O to form KF occurs. At 900 °C, CaF_2_ tends to vaporize; when the temperature range is 1050–1500 °C, the evaporation ratio of CaF_2_ exceeds that of KF. The evaporation proportion of KF at each temperature point rises with the addition of K_2_O to the slag. MgF_2_ begins to evaporate at 900 °C; from 1350 °C, CaF_2_ reacts with Al_2_O_3_ to form AlF_3_ and AlOF. When compared with CaF_2_ and KF, the evaporation of MgF_2_, AlF_3_, and AlOF is negligible. When both K_2_O and Na_2_O are contained in the slag, CaF_2_, KF and NaF are the main evaporated substances. Both KF and NaF begin to evaporate from 650 °C. The evaporation of KF is gradually reduced, while the evaporation of NaF first increases and then decreases. In any case, at the beginning of the reaction, the vaporized gas is almost entirely KF; this is because K_2_O and CaF_2_ can react violently at a low temperature of 650 °C, while other oxides must reach a certain temperature in order to react violently.

#### 3.1.2. Vapor Pressure

The makeup of the slag system has a direct impact on the evaporation of fluoride. When K_2_O and Na_2_O are added to the slag system, the following chemical reactions will occur, as shown in Equations (1)–(6).
CaF_2_ (s) = CaF_2_ (g) (1)
K_2_O (s) + CaF_2_ (s) = 2 KF (g) + CaO (s) (2)
Na_2_O (s) + CaF_2_ (s) = 2 NaF (g) + CaO (s) (3)
CaF_2_ (s) + MgO (s) = CaO (s) + MgF_2_ (g) (4)
3 CaF_2_ (s) + Al_2_O_3_ (s) = 3 CaO (s) + 2 AlF_3_ (g) (5)
CaF_2_ (s) + Al_2_O_3_ (s) = CaO (s) + 2 AlOF (g)(6)

The thermodynamics in the previous section calculated that CaF_2_, KF, and NaF are the main evaporation products, and the generated MgF_2_, AlF_3_ and AlOF evaporation gases are lesser and negligible. Thus, Equations (1)–(3) are anticipated to play a major part in the evaporation process.

The propensity of the molecules in solution to separate from the slag and enter space is reflected in the vapor pressure of fluoride gas, a tendency also known as escape tendency. The equilibrium constants of chemical Equations (1)–(3) can be expressed as follows:(7)$$K_{\mathrm{Eq}.(1)}=\frac{P_{{\mathrm{CaF}}_{2}}}{a_{{(\mathrm{CaF}}_{2})}}$$
(8)$$K_{\mathrm{Eq}.(2)}=\frac{a_{\left(\mathrm{CaO}\right)}\cdot P_{\left(\mathrm{KF}\right)}^{2}}{a_{{(\mathrm{CaF}}_{2})}\cdot a_{{(K}_{2}O)}}$$
(9)$$K_{\mathrm{Eq}.(3)}=\frac{a_{\left(\mathrm{CaO}\right)}\cdot P_{\left(\mathrm{NaF}\right)}^{2}}{a_{{(\mathrm{CaF}}_{2})}\cdot a_{{(\mathrm{Na}}_{2}O)}}$$

Here, *P_i_* represents the vapor pressures (atm) of melt component *i* and *a_i_* is the activity of the melt component *i* in the molten slags. The FactSage 8.0 software was used to calculate the activities and equilibrium constants of CaF_2_, Na_2_O, and K_2_O with respect to the pure liquid standard state and CaO with respect to the pure solid standard state. According to calculations, the equilibrium constants for Equations (1)–(3) are 4.45 × 10^−4^ [26], 1.10 × 10^5^ [27], and 1.04 × 10^2^ [28], respectively. Table 2 provides the data for the activity calculations.

By inputting the equilibrium constant values and activity data into Equations (7)–(9), the vapor pressures of CaF_2_, KF, and NaF are determined. The outcomes are shown in Table 3 and Table 4.

When there are various vapor pressures in a gas mixture, the component with the greater vapor pressure will evaporate first. KF has a greater vapor pressure than CaF_2_ at 1500 °C; this indicates that KF is more likely to evaporate at this temperature. When both K_2_O and Na_2_O are added to the slag sample, KF is always in the leading position, and its evaporation ability is stronger than that of CaF_2_ and NaF.

### 3.2. TG and DSC Analysis of the Slag

Two experiments with the same conditions were carried out on K3 slag. Dotted lines depict the outcomes of the second experiment. The curves in Figure 3 almost overlap, indicating that the experiment is reproducible. TG and DSC experimental results of K_2_O content are summarized in Figure 4. It should be observed that when K_2_O content steadily increases from 0% to 8.3 wt%, evaporation increases. When the K_2_O content in the slag is 0, 2.3, 5.3, and 8.3 wt%, the evaporation rates of the slag are 0.76%, 0.93%, 1.09%, and 1.21%, respectively. The evaporation is greatest at 8.3 wt% of K_2_O and lowest without K_2_O. K_2_O plays a role in promoting evaporation in the slag system.

The thermogravimetric curves of K1, K2, and K3 show a sudden and violent weight loss at around 650 °C. This is likely due to the beginning of the reaction between K_2_O and CaF_2_ to form KF; this can be verified using the thermodynamic calculations shown in Section 3.1. A second violent weight loss occurs at 1200–1500 °C. As can be seen in Figure 2, a large amount of CaF_2_ gas is generated at this stage, as along with a small amount of KF, MgF_2_, AlF_3_, and AlOF. This process involves Equations (1), (2) and (4)–(6).

The DSC curves in dashed lines are shown in Figure 4. In general, this process can be broken down into two phases: (I) the glass transition period, which corresponds to the exothermic peak, indicating the event that amorphous glassy phase transforms to crystalline state during the heating process, and (II) the melting period, which corresponds to two endothermic peaks [29]. The melting stage is primarily the evaporation behavior of fluoride. The first endothermic peak at this stage clearly shows that only A slag without K_2_O is not formed. This endothermic peak represents the continuous reaction of K_2_O and CaF_2_ to form KF; KF is still escaping during this process. The second endothermic peak in the melting stage represents the evaporation of CaF_2_ itself and the reaction of Al_2_O_3_ and MgO with CaF_2_. The endothermic peak of A slag shifts to a higher temperature, showing that A slag requires a higher temperature to react, thus indicating that K_2_O can reduce the melting temperature of the slag.

Figure 5 presents the TG and DSC curves of samples K4 and K5. The evaporation rates of samples K4 and K5 are 1.48% and 1.32%, respectively. By comparing the TG diagrams of K4 and K5 slag, it can be understood that under the premise that the total amount of K_2_O and Na_2_O added is equal, when the amount of K_2_O added is more than Na_2_O, the amount of evaporation is relatively large, indicating that K_2_O has a greater impact on evaporation. The DSC curve’s exothermic peak denotes the transition from the amorphous to the crystalline state. The transformation temperature and peak intensity of K5 slag are higher than those of K4 slag; this shows that the transformation driving force of K4 slag is larger.

### 3.3. FTIR Spectra of the Slag Sample

The FTIR spectra of the CaF_2_-CaO-Al_2_O_3_-MgO-TiO_2_-(Na_2_O-K_2_O) slag system as a function of wavenumber is shown in Figure 6. The spectral bands can be divided into three ranges: 930–660 cm^−1^, 660–550 cm^−1^, and 550–450 cm^−1^, which correspond to the asymmetric stretching vibration of the (AlO*_n_*F_4−*n*_)-tetrahedral complexes (*n* = 0–4), (AlO_4_)-tetrahedra, and (AlO_6_)-octahedra, respectively. Moreover, the Ti-O stretching vibration is seen at a wave number of around 425 cm^−1^ [30].

The transmission band of the (AlO*_n_*F_4−*n*_)-tetrahedral complexes exists in the fluoroaluminate system. The existence of (AlO*_n_*F_4−*n*_)-tetrahedral complexes is based on the increased vibrational asymmetry caused by the coexistence of Al-F and Al-O bonds in the complexes [31]. As K_2_O content increases, the gravity center of the (AlO*_n_*F_4−*n*_)-tetrahedral complexes shifts to lower wavenumbers. This transition means that the distance between Al and O widens, that is, increasing the K_2_O content can depolymerize the network structure in the molten slag [32]. In other words, the degree of polymerization of (AlO*_n_*F_4−*n*_)-tetrahedral complexes decreases with the increase in K_2_O content. (AlO_4_)-tetrahedra behaves as network formers. At a wave number of about 625 cm^−1^, a trough appears in the (AlO_4_)-tetrahedra; the trough is even smaller in the absence of K_2_O addition. The Al-O bond in the (AlO_4_)-tetrahedral structure has a gentler slope at 620 cm^−1^; this indicates a more complex network structure when K_2_O is not added. When K_2_O, a strong alkali oxide, is added, free oxygen ions (O^2−^) can be produced in the slag. O^2−^ can break the Al-O bond of the (AlO_4_)-tetrahedral structure, depolymerizing the network complex structure of aluminate into simpler structural units. The center of gravity of (AlO_6_)-octahedra shifts to a higher wave number, and the transmission peak intensity of (AlO_6_)-octahedra increases relatively. According to previous studies [33], this is because O^2−^ provided by K_2_O reacts with (AlO_4_)-tetrahedra to produce (AlO_6_)-octahedra, thus promoting the depolymerization of slag network structure. This is related to the Ti-O stretching vibration at about 420 cm^−1^, and varying K_2_O content has a slight effect on the structure. This demonstrates that the Ti-O bond in the titanate structure is less affected by increasing the K_2_O level.

Through the above structural analysis, the addition of K_2_O to the slag significantly depolymerizes (AlO*_n_*F_4−*n*_)-tetrahedral complexes, (AlO_4_)-tetrahedra, and (AlO_6_)-octahedra network structures, but has little effect on the stretching vibration of Ti-O. It further explains the TG curves of the previous section, where evaporation intensifies with increasing K_2_O content.

Figure 7 compares the structural effects of Na_2_O and K_2_O on the slag. The spectrum of the current slag system shows four visible transmittance peaks, which corresponds to asymmetric stretching vibration of the (AlO*_n_*F_4−*n*_)-tetrahedral complexes (*n* = 0–4) in the wavenumbers of 900–680 cm^−1^, the (AlO_4_)-tetrahedra in the wavenumbers of 680–600 cm^−1^, the (AlO_6_)-octahedra in the wavenumbers of 600–490 cm^−1^, and the stretching vibration of Ti-O at 420 cm^−1^.

It can be seen in Figure 7 that the (AlO*_n_*F_4−*n*_)-tetrahedral complexes’ center of gravity changes to higher wavenumbers; this indicates that the addition of K_2_O may result in a lesser degree of polymerization of the network structure in the slag than that of Na_2_O. In the (AlO_4_)-tetrahedra wavenumber range, the K5 slag with a greater Na_2_O content (about 660 cm^−1^) exhibits a smaller slope, indicating that adding more Na_2_O will show a more complex network structure. The addition of K_2_O and Na_2_O has no obvious effect on the transmission band of the (AlO_6_)-octahedra. The stretching vibration of Ti-O around 420 cm^−1^ has no obvious change, indicating that the addition of K_2_O or Na_2_O has little effect on the Ti-O bond in the titanate structure.

Based on the aforementioned structural analysis, when the total amount of K_2_O and Na_2_O is the same, slag containing more K_2_O can better depolymerize (AlO*_n_*F_4−*n*_)-tetrahedral complexes and (AlO_4_)-tetrahedra network structures, and has little effect on (AlO_6_)-octahedra and Ti-O stretching vibration. This indicates that the K^+^ provided by K_2_O is more advantageous than the Na^+^ provided by Na_2_O. This can also explain why the evaporation of K_2_O is more intense for the same total amount of Na_2_O and K_2_O added.

## 4. Conclusions

When the slag contains both K_2_O and Na_2_O, the main evaporating substances are CaF_2_, KF, and NaF. In comparison, MgF_2_, AlF_3_, and AlOF seldom ever evaporate. KF and NaF begin to evaporate at 650 °C. NaF increases and subsequently decreases as the temperature rises, while KF decreases as temperature rises. At 900 °C, CaF_2_ tends to evaporate, and the evaporation intensifies with the increase of temperature. At the beginning of the reaction, KF dominates absolutely, while CaF_2_ dominates when it exceeds 1050 °C.The vapor pressure of KF is stronger than that of CaF_2_ at 1500 °C. When K_2_O and Na_2_O are added to the residue sample at the same time, the evaporation ability of KF is stronger than CaF_2_ and NaF.Evaporation increases from 0.76% to 1.21% when K_2_O content rises from 0% to 8.3 wt%. The evaporation rates of samples K4 and K5 are 1.48% and 1.32%, respectively. Under the premise that the total amount of K_2_O and Na_2_O added is equal, when the amount of K_2_O added is greater than Na_2_O, the evaporation rate is relatively large, indicating that K_2_O has a significant influence on evaporation.FTIR results show that with the addition of K_2_O, the (AlO*_n_*F_4−*n*_)-tetrahedral complexes, (AlO_4_)-tetrahedra, and (AlO_6_)-octahedra network structures are depolymerized; this has little effect on the stretching vibration of Ti-O. Comparing the effects of Na_2_O and K_2_O addition under the same conditions, it is found that the slag with higher K_2_O content can better depolymerize the (AlO*_n_*F_4−*n*_)-tetrahedral complexes and (AlO_4_)-tetrahedra network structures, and has little effect on the (AlO_6_)-octahedra and Ti-O stretching vibration.Although the addition of a small amount of alkali metals can promote the partial evaporation of slag, it can also change the melting characteristics of the ESR slag system, including viscosity and melting temperature, which are conducive to the melting of slag system. The applicability of these characteristics in industry should be considered comprehensively.

## Figures and Tables

**Figure 1 materials-16-02777-f001:**
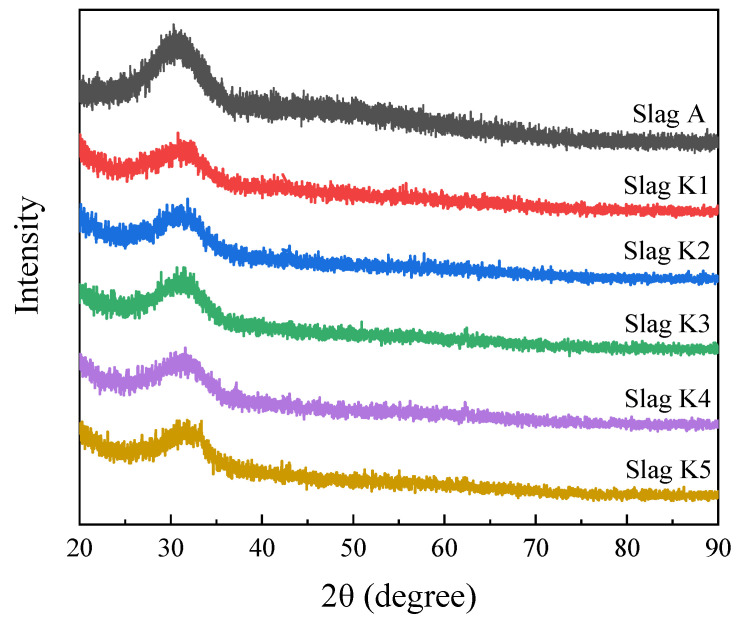
The XRD patterns of the pre-melted slags.

**Figure 2 materials-16-02777-f002:**
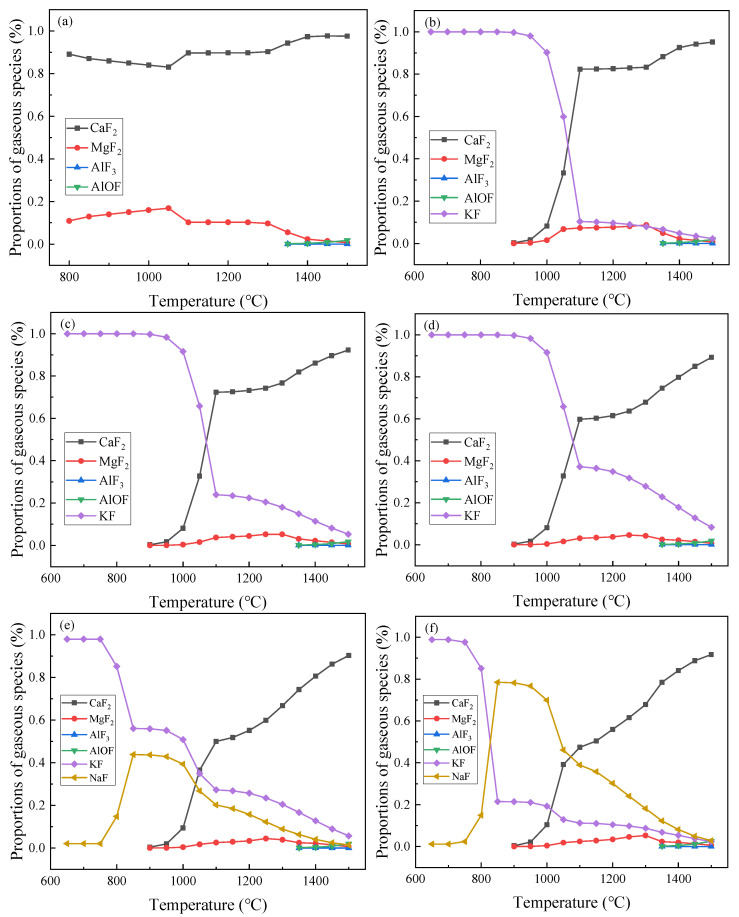
The proportions of main fluorine gases at different temperatures: (**a**) A, (**b**) K1, (**c**) K2, (**d**) K3, (**e**) K4, (**f**) K5.

**Figure 3 materials-16-02777-f003:**
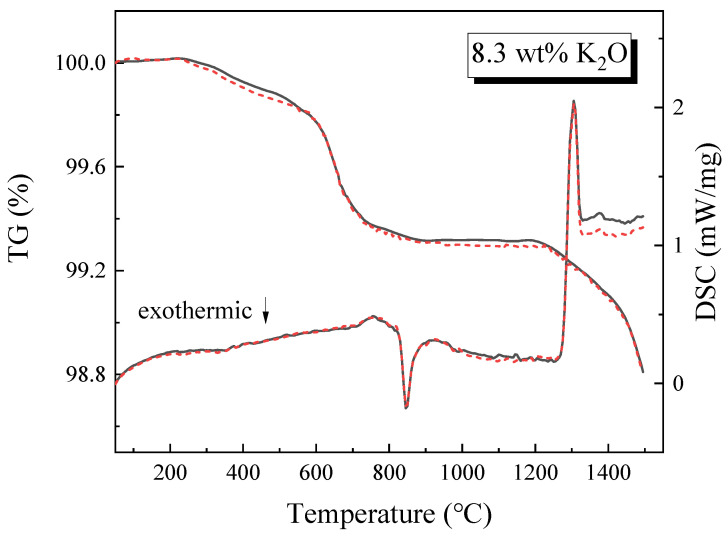
The repeatability of K3 slag at a heating rate of 20 °C/min and an argon flow rate of 80 mL/min: first experiment (solid line); the second experiment (dash line).

**Figure 4 materials-16-02777-f004:**
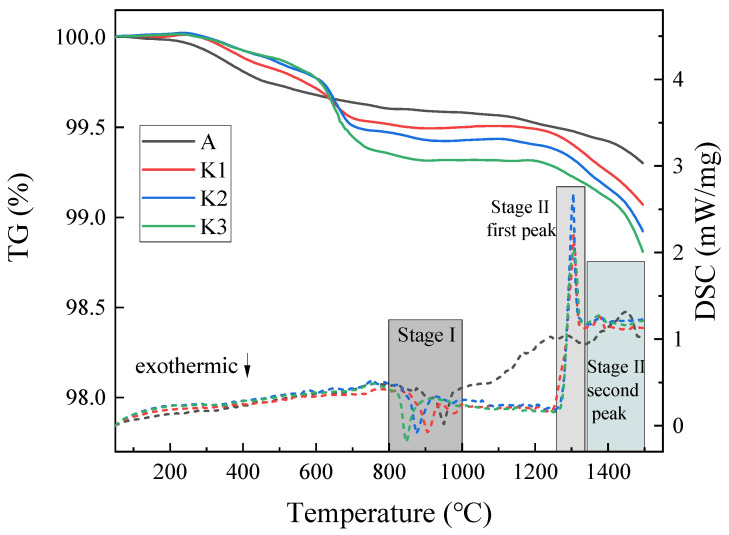
The TG (solid line) and DSC (dotted line) curves of the slag at different K_2_O contents.

**Figure 5 materials-16-02777-f005:**
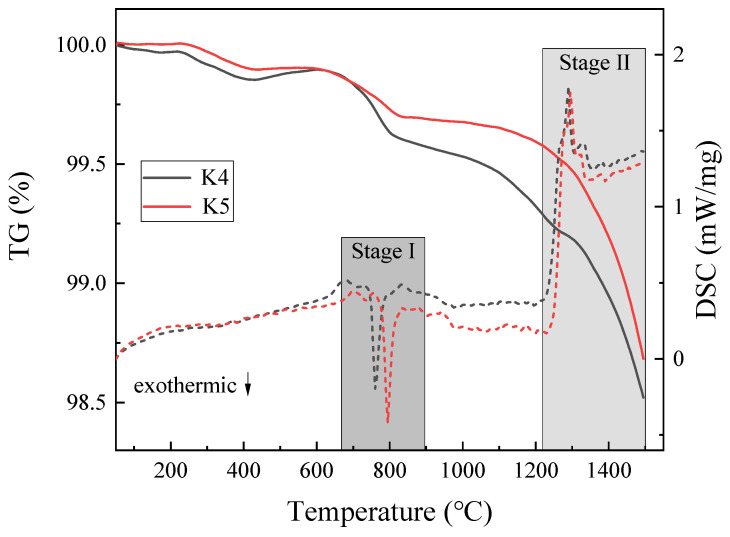
The TG (solid line) and DSC (dotted line) curves of the slag K4 and K5.

**Figure 6 materials-16-02777-f006:**
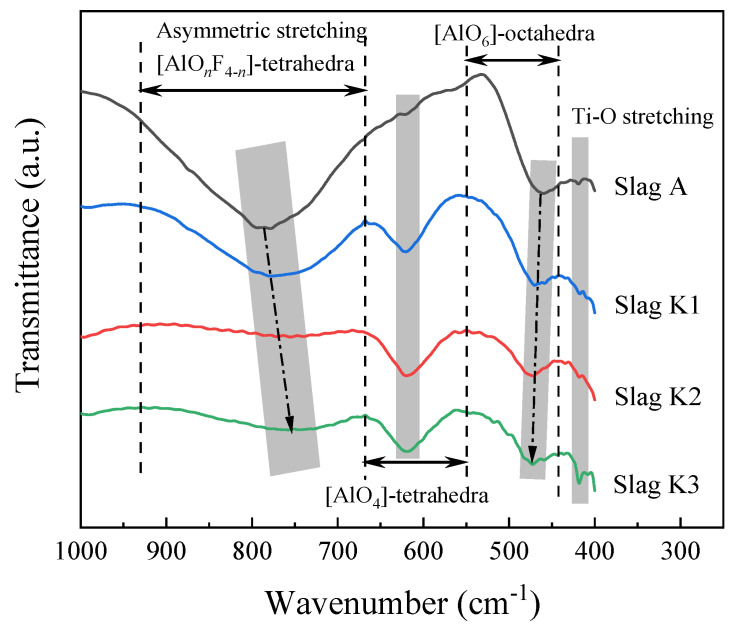
FTIR spectra for the slags with different K_2_O content: slag A, 0 wt% K_2_O; slag K1, 2.3 wt% K_2_O; slag K2, 5.3 wt% K_2_O; slag K3, 8.3 wt% K_2_O.

**Figure 7 materials-16-02777-f007:**
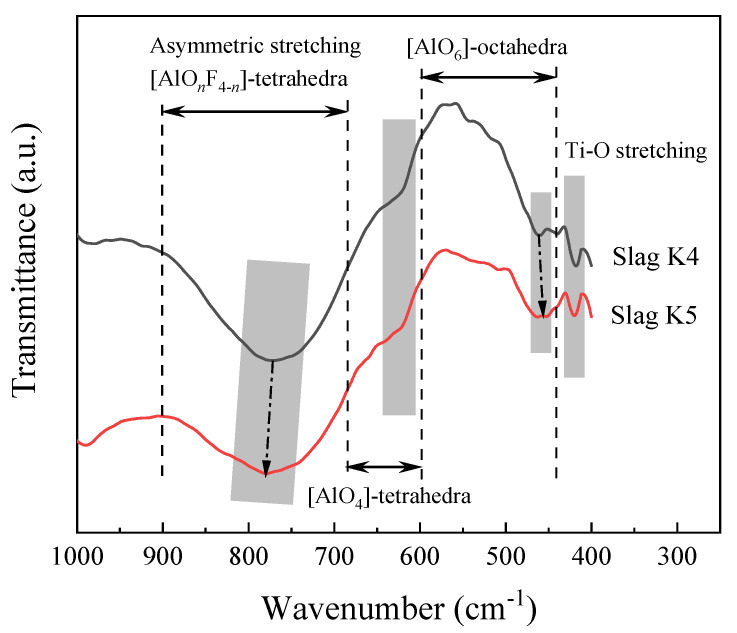
FTIR spectra for the slags of K4 and K5.

**Table 1 materials-16-02777-t001:** Chemical compositions of the slag samples (wt%).

Slag	Pre-Experimental Composition (Designed)							Post-Experimental Composition (XRF)						
	CaF_2_	CaO	Al_2_O_3_	MgO	TiO_2_	Na_2_O	K_2_O	CaF_2_	CaO	Al_2_O_3_	MgO	TiO_2_	Na_2_O	K_2_O
A	30.0	30.0	30.0	2.0	8.0	0.0	0.0	29.8	31.2	29.5	1.8	7.7	0.0	0.0
K1	30.0	28.5	28.5	2.0	8.0	0.0	3.0	29.7	29.8	29.2	1.7	7.3	0.0	2.3
K2	30.0	27.0	27.0	2.0	8.0	0.0	6.0	27.9	29.9	27.3	1.8	7.8	0.0	5.3
K3	30.0	25.5	25.5	2.0	8.0	0.0	9.0	26.8	28.9	26.2	1.9	7.9	0.0	8.3
K4	30.0	25.5	25.5	2.0	8.0	3.0	6.0	25.9	29.5	26.5	1.9	7.3	3.2	5.7
K5	30.0	25.5	25.5	2.0	8.0	6.0	3.0	25.6	29.7	27.3	1.7	7.2	6.3	2.2

**Table 2 materials-16-02777-t002:** The activity of each component at 1500 °C using FactSage 8.0 software.

	CaF_2_	KF	NaF
A	3.86 × 10^−4^	-	-
K1	3.89 × 10^−4^	4.75 × 10^−2^	-
K2	3.90 × 10^−4^	6.18 × 10^−2^	-
K3	3.92 × 10^−4^	6.21 × 10^−2^	-
K4	3.92 × 10^−4^	4.87 × 10^−2^	1.15 × 10^−2^
K5	3.93 × 10^−4^	3.09 × 10^−2^	2.76 × 10^−2^

**Table 3 materials-16-02777-t003:** The vapor pressure of A-K3 calculated at 1500 °C using FactSage 8.0 software.

	CaF_2_	KF
A	3.86 × 10^−4^	-
K1	3.89 × 10^−4^	4.75 × 10^−2^
K2	3.90 × 10^−4^	6.18 × 10^−2^
K3	3.92 × 10^−4^	6.21 × 10^−2^

**Table 4 materials-16-02777-t004:** The vapor pressure of K4 and K5 calculated at 1500 °C using FactSage 8.0 software.

	CaF_2_	KF	NaF
K4	3.92 × 10^−4^	4.87 × 10^−2^	1.15 × 10^−2^
K5	3.93 × 10^−4^	3.09 × 10^−2^	2.76 × 10^−2^

## Data Availability

Not applicable.

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
