# Peer review of "Fluoride Evaporation of Low-Fluoride CaF2-CaO-Al2O3-MgO-TiO2-(Na2O-K2O) Slag for Electroslag Remelting"

_materials, 2023, doi:10.3390/ma16072777_

Round 1

Reviewer 1 Report

Submitted article Fluoride evaporation of low-fluoride CaF2-CaO-Al2O3-MgO- 2TiO2-(Na2O-K2O) slag for electroslag remelting is presented at a quality level and the results are clearly presented. I recommend publishing the article in the presented form.

Author Response

Point 1: Submitted article Fluoride evaporation of low-fluoride CaF2-CaO-Al2O3-MgO-TiO2-(Na2O-K2O) slag for electroslag remelting is presented at a quality level and the results are clearly presented. I recommend publishing the article in the presented form.

Response:  Thank you for your affirmation. The changes made to the manuscript can be seen in the attachment.

Reviewer 2 Report

Authors have done much work determining the effect of adding alkali metal oxides to electroslag remelting slag on the type and amount of components evaporated at high temperatures.

Besides having low melting points both these oxides and their fluorides can evaporate even at slag premelting even before the ESR process. The author’s data also have shown that both K2O and Na2O addition in ESR slag increases evaporation of fluorides that is negative, especially for low fluoride slags because it will reduce their electric conductivity. The article [19], which the author used to ground their idea to use K2O and Na2O additions to ESR slags, describes another case – fluoride-less slag for mould casting. All previous experience reports rather a negative effect of alkali additions to ESR slags (as own as that can be found in books). These oxides are not good for ESR slag because working temperature is too high (1650-1700 C according to different estimations).

When we are talking about ESR slags three characteristics are especially important: melting temperature interval, electric conductivity and viscosity. To make any conclusion about ESR slags with proposed additions, it is necessary to measure these properties. However, it is already understood that if evaporations occur and change slag properties, slag properties change while remelting (going for many hours), and the ESR process will be violated by slag composition changing.

The authors must write their conclusion about the usability of investigated slags at electroslag remelting.

If authors think that research belongs to physical chemistry rather than metallurgy, it is possible to remove ESR as such a slag application. In another case, this work gives no valuable data because this slag is not suitable for the ESR process.

Author Response

Point 1: Authors have done much work determining the effect of adding alkali metal oxides to electroslag remelting slag on the type and amount of components evaporated at high temperatures.

Besides having low melting points both these oxides and their fluorides can evaporate even at slag premelting even before the ESR process. The author’s data also have shown that both K2O and Na2O addition in ESR slag increases evaporation of fluorides that is negative, especially for low fluoride slags because it will reduce their electric conductivity. The article [19], which the author used to ground their idea to use K2O and Na2O additions to ESR slags, describes another case-fluoride-less slag for mould casting. All previous experience reports rather a negative effect of alkali additions to ESR slags (as own as that can be found in books). These oxides are not good for ESR slag because working temperature is too high (1650-1700 C according to different estimations).

When we are talking about ESR slags three characteristics are especially important: melting temperature interval, electric conductivity and viscosity. To make any conclusion about ESR slags with proposed additions, it is necessary to measure these properties. However, it is already understood that if evaporations occur and change slag properties, slag properties change while remelting (going for many hours), and the ESR process will be violated by slag composition changing.

The authors must write their conclusion about the usability of investigated slags at electroslag remelting.

If authors think that research belongs to physical chemistry rather than metallurgy, it is possible to remove ESR as such a slag application. In another case, this work gives no valuable data because this slag is not suitable for the ESR process.

Response: Thank you for your suggestion. According to a previous study by Ju et al.[1], when Na2O, an alkaline oxide, was added, the viscosity of the slag was reduced and the melting point of the slag system was lowered, suggesting that Na2O can be used as a replacement for CaF2 in fluorinated slag systems for the development of electroslag remelting slag systems. Regarding the addition of K2O to this slag system, an experimental study was also conducted during the warming process, and it was concluded that the addition of K2O could lower the melting point as well as the viscosity of the slag system, and enhance the conductivity for the electroslag remelting process, as reflected in another unpublished manuscript. The corresponding conclusions have been added in this manuscript. Please see the attachment.

[1] Ju, J.; Yang, K.; Gu, Y.; He, Kun. Effect of Na2O on Viscosity, Structure and Crystallization of CaF2-CaO-Al2O3-MgO-TiO2 Slag in Electroslag Remelting. Russ. J. Non-Ferrous Met. 2022, 63, 599-609. doi: 10.3103/S1067821222060098.

Reviewer 3 Report

The authors must make small changes in the article.

- The introduction must be detailed.
- The authors must highlight the degree of originality of their research.
- Table 1 must appear completely (it is truncated - a part on the right is missing).
- In Figures 6 and 7, the text must be redone - there are terms that are not correct.

Author Response

The authors must make small changes in the article.

Point 1: The introduction must be detailed.

Response: Thank you for your suggestion. Additions have been made in the introduction. Please see the attachment.

Point 2: The authors must highlight the degree of originality of their research.

Response: Thank you for your suggestion. The degree of originality of this manuscript has been highlighted. Please see the attachment.

Point 3: Table 1 must appear completely (it is truncated-a part on the right is missing).

Response: I apologize for not paying attention to the table formatting and have made changes in the original manuscript. Please see the attachment.

Point 4: In Figures 6 and 7, the text must be redone-there are terms that are not correct.

Response: Thank you for your suggestion. The relevant content has been corrected in the manuscript. Please see the attachment.

Reviewer 4 Report

The authors studied the non iso thermal process evaporation of the low-fluoride CaF2-CaO-Al2O3-MgO-TiO2-(Na2o-K2O) slag. The analysis of variation of the slag structure by FTIR is quiet interesting and relevant to the researchers on electroslag remelting process. The paper is well writtenand easy to understand, The conclusions arrived are well consistent with the objectives of the research.

1. The authors claim that the increase of K2O in the slag can depolymerize better than Na2O. The mechanism behind this may be included with the help of TG/DSC experiments.

Author Response

The authors studied the non iso thermal process evaporation of the low-fluoride CaF2-CaO-Al2O3-MgO-TiO2-(Na2O-K2O) slag. The analysis of variation of the slag structure by FTIR is quiet interesting and relevant to the researchers on electroslag remelting process. The paper is well written and easy to understand, the conclusions arrived are well consistent with the objectives of the research.

Point 1: The authors claim that the increase of K2O in the slag can depolymerize better than Na2O. The mechanism behind this may be included with the help of TG/DSC experiments.

Response: Thank you for your suggestion. The relevant descriptions have been added in the manuscript. Please see the attachment.

Round 2

Reviewer 2 Report

Very unfortunately authors didn't react on my questions and suggestions. I still be sure that the authors have not shown the applicability of the studied slags to the ESR process (see my questions in previous review). The fact that this is true very easy to check if you specify the search for the words "electroslag remelting". They are only in the introduction and references. Added reference onto own publication is not enough convincing

Author Response

Point 1: Very unfortunately authors didn't react on my questions and suggestions. I still be sure that the authors have not shown the applicability of the studied slags to the ESR process (see my questions in previous review). The fact that this is true very easy to check if you specify the search for the words "electroslag remelting". They are only in the introduction and references. Added reference onto own publication is not enough convincing.

Response: Thank you for your rigorous academic style. I am sorry for my misunderstanding before. The high melting point and high viscosity of the low-fluorine slag system have a great impact on the smelting process. The introduction to the manuscript has added support from other scholars' studies on the addition of alkali metal oxides to electroslag remelting slag systems and concluded that the addition of alkali metal is applicable to the development of electroslag remelting slag systems. Studies in the literature show that the addition of alkali metal is beneficial to solve the above problems, but the effect of alkali metal addition on evaporation is less studied. In this paper, based on the previous studies, a laboratory study of evaporation is conducted to predict the feasibility of adding alkali metal K2O to the electroslag remelting slag systems. Please see the attachment.
